# The Combined Use of Gentamicin and Silver Nitrate in Bone Cement for a Synergistic and Extended Antibiotic Action against Gram-Positive and Gram-Negative Bacteria

**DOI:** 10.3390/ma14123413

**Published:** 2021-06-20

**Authors:** John Jackson, Joey Lo, Eric Hsu, Helen M. Burt, Ali Shademani, Dirk Lange

**Affiliations:** 1Faculty of Pharmaceutical Sciences, University of British Columbia, Vancouver, BC V6T 1Z3, Canada; etjh111@gmail.com (E.H.); helen.burt@ubc.ca (H.M.B.); 2The Stone Centre at Vancouver General Hospital, Department of Urologic Sciences, University of British Columbia, Vancouver, BC V6T 1Z3, Canada; jlo@prostatecentre.com (J.L.); dirk.lange@ubc.ca (D.L.); 3Department of Biomedical Engineering, University of British Columbia, Vancouver, BC V6T 1Z3, Canada; ali.shademani@mech.ubc.ca

**Keywords:** bone cement, gentamicin, silver, antibacterial synergy

## Abstract

Using bone cement as a carrier, gentamicin was for years the default drug to locally treat orthopedic infections but has lost favor due to increasing bacterial resistance to this drug. The objective of this study was to investigate the effect of combining gentamicin with silver nitrate in bone cement against *S. aureus* and *P. aeruginosa*. Antibacterial effects (CFU counts) of gentamicin and silver were initially studied followed by studies using subtherapeutic concentrations of each in combination. The release rates from cement were measured over 10 days and day 7 release samples were saved and analyzed for antibiotic activity. A strong synergistic effect of combining silver with gentamicin was found using both dissolved drugs and using day 7 bone cement release media for both Gram-positive and Gram-negative bacteria. The cement studies were extended to vancomycin and tobramycin, which are also used in bone cement, and similar synergistic effects were found for day 7 release media with *P. aeruginosa* but not *S. aureus*. These studies conclude that the combined use of low loadings of gentamicin and silver nitrate in bone cement may offer an economical and much improved synergistic method of providing anti-infective orthopedic treatments in the clinic.

## 1. Introduction

Approximately one million hip and knee replacement surgeries are completed each year in the USA. Despite improved surgical methods and materials, the rate of infection remains at up to 2% [1]. While this may seem low, the impact is quite significant as the treatment for such infections is an expensive two-stage revision arthroplasty whereby the implant is removed, a polymeric spacer is placed in the site to maintain the void and the patient is treated with antibiotics [1,2,3,4,5]. Only once the infection has been treated completely is the original replacement surgery repeated with a new implant. While the main function of the bone cement (poly methyl methacrylate (PMMA)) spacer is to maintain the void for subsequent implantation of the new implant, its implantation into the affected area makes it a convenient way to deliver antibiotics directly into the infected environment to assist the clearance of the infection [6]. For this purpose, PMMA spacers contain an antibiotic that is slowly released over the 3-month period the spacer is in place.

For many years, gentamicin was the antibiotic of choice to treat orthopedic implant infections mainly due to it being inexpensive and has a broad range of activity against both Gram-positive and Gram-negative bacterial species. Given the significant increase in resistance towards gentamycin by many bacterial species including Meticillin-resistant *Staphylococcus aureus* (MRSA) [2,5,7], the field has moved towards the use of tobramycin or vancomycin that are encapsulated in the spacers [2,3,8]. That said, recent reports of bacterial resistance towards tobramycin among species associated with orthopedic implants [8,9] has made vancomycin the primary drug against MRSA in the orthopedic settings. Despite still being effective against MRSA, some resistance to vancomycin is already being reported resulting for the need for alternative approaches to the treatment and prevention of orthopedic implant-associated infections [10,11,12].

Aside from the obvious challenge of antibiotic resistance, another issue that needs to be improved is the slow release of drugs from PMMA, attributable to its non-porous nature, resulting in a small burst phase followed by the release of minute and often subtherapeutic amounts of drug resulting in less than 10% of the loaded drug being released over a 1–2 month period in aqueous media [8,13,14,15,16]. While minimal drug release is not very effective at killing the infecting bacteria, exposure to subtherapeutic levels of the drug also contribute to the development of resistant species significantly complicating subsequent infections [17]. To overcome these release profile problems many groups have incorporated hydrophilic excipients such as acrylic acid, xylitol, or cyclodextrin into the bone cement to increase porosity and release deeper depots of the drug [17,18,19]. While partially effective, the inclusion of excess drugs and release excipients was shown to have adverse effects on the mechanical strength of PMMA. 

Previous work by our group has shown that the inclusion of large amounts (up to 18% *w/w*) of sodium chloride or dextran in PMMA resulted in significant increases in the burst phase and extended release rates of vancomycin, linezolid or fusidic acid from bone cement with only marginal effects on mechanical strength [20]. While this approach certainly helps to overcome the drug release issue, it still relies on single drugs to which resistance is on the rise significantly impacting the treatment of serious orthopedic implant-associated infections. An alternative approach to this may be the inclusion of an antimicrobial cocktail whereby an additive or synergistic mode of action might override the mechanism of resistance and allow the primary drug to treat such bacteria.

Silver is currently used as an anti-infective agent in numerous wound dressing materials (e.g., Acticoat ^TM^) and as coatings on catheter lines (e.g., Bardex). Although traditionally used as a silver salt, improved activity and decreased tissue affect have been noted when it is used in a nanoparticle form. Previous studies have shown silver-loaded PMMA to outperform gentamicin in vivo [21]. Similarly, PMMA loaded with silver nanoparticles was shown to be effective against resistant *S. aureus* and *S. epidermidis,* while 2% gentamicin was not [22]. Similarly, Wekwejt et al. showed superior inhibition of colonization and biofilm formation by both Gram-positive and Gram-negative bacteria on bone cement loaded with silver nanoparticles compared to gentamycin loaded cement [23]. These observations along with the need to develop alternative solutions to overcome the negative effects of antibiotic resistance on the treatment and prevention of orthopedic implant-associated infections has led groups to investigate the use of silver (usually as nanoparticles) in combination with gentamycin in bone cement [23,24] against orthopedically relevant bacteria. Perhaps the most compelling argument for the use of silver-based approaches is the synergistic effects with many commonly used antibiotics against both Gram-positive and Gram-negative bacteria [25,26,27]. The mechanism of action of silver as an antibiotic is unclear and seems to involve many processes whereby silver ions bind to sulphydryl groups on proteins and disrupt function, bind to DNA and RNA to inhibit replication, and induced membrane damage in bacteria [24]. Silver nanoparticles bind to bacterial membranes to perturb permeability and membrane protein function as well as entering the cell to inhibit respiratory chain processes [26,28,29,30,31].

The objective of the present study was to investigate any additive or synergistic antibiotic effects of combining gentamicin with silver nitrate on *S. aureus* and *P. aeruginosa* two common pathogens associated with complicated orthopedic implant infections. Release profiles from bone cement and antibacterial activity were assessed as part of this proof of concept study. In light of the good results observed for gentamicin in combination with silver nitrate, the studies were extended to included two other commonly used antibiotics in bone cement, tobramycin or vancomycin in combination with silver nitrate.

## 2. Materials and Methods

Bone cement (Palacos) with and without gentamicin was obtained from Heraeus Medical (Hanau, Germany). Gentamicin sulfate, vancomycin, tobramycin, DextranT70 and silver nitrate (all 99% purity) were obtained from Sigma-Aldrich (St. Louis, MO, USA). Fluoraldehyde-O-pthaldehyde was obtained from Thermo Fisher, Burnaby, BC, Canada. Luria Bertani broth was obtained from Gibco—Thermo Fisher, Burnaby, BC, Canada. 

### 2.1. Bone Cement Film Manufacture

Bone cement kits contain polymethyl methacrylate powder and a vial of liquid methyl methacrylate monomer. When added together at a 2:1 weight ratio, the powder becomes a very viscous paste which may be molded into any form. The material must be handled quickly as is sets within 15 min. In these studies the paste was compressed between two glass slides to form a thin film of cement paste and any excess pastes that squeezed out the sides was removed. 

1.5 g films were made as previously described [20] by vigorous blending of 1 g of Palacos powder with 0.5 g of the liquid methyl methacrylate (both contained in the kit). The paste was quickly spread evenly over a glass microscope slide (25 mm × 75 mm × 1 mm) and a second microscope slide was applied sandwich style and the paste compressed to form a uniform bone cement layer 1 mm thick. After 2 h the microscope slides were teased apart gently using a scalpel and the thin film of bone cement was freed from the glass surface. The thickness was checked using a digital micrometer. In some cases silver nitrate was added by preblending the material (19 mg to give 0.8% silver *w/w* in the final paste) with the dry Palacos powder prior to mixing. When dextran was added at 18% *w/w* to the Palacos powder then 270 mg of dextran was added to 730 mg of Palacos cement for a total weight of 1 g then 0.5 g of methyl methacrylate was added to make the paste. For tobramycin or vancomycin loaded pastes the same methods were used but dry drug (12 mg) was blended into the Palacos cement powder (988 mg) at 0.8% *w/w* (final after 0.5 g of methyl methacrylate added).

### 2.2. Drug Release Studies

Bone cement samples that had been cast as thin films were cut into 250 mg pieces (*n* = 4) and placed in 20 mL glass capped vials (Fisher, Burnaby, BC, Canada). Release media (1.5 mL of Hepes buffer (0.01 M) pH 7.4) was added and the vials placed in a 37 °C incubator. At time points of 2 h, 6 h, 1, 2, 4, 7 and 10 days, all the release media was collected, stored frozen and replaced with 1.5 mL of fresh media. Four samples of bone cement in vials per condition were set up and one sample was taken from each vial per time point.

#### Method of Drug Analysis

Gentamicin release was measured using a one-step quantitative fluorescence assay. Firstly a calibration curve was established for the drug as follows: A fresh stock solution of gentamicin was made up in water at 100 µg/mL and serial diluted in water to give a set of calibration standards. To 100 µL of each calibration standard was added 100 µL of fluoraldehyde-O-pthaldehyde using a 96 well cell culture plate and left for 5 min. The plate was then read in a plate fluorimeter (Thermo fluorskan) using excitation and emission wavelengths filters of 355 and 450 nm. Calibration graphs were linear in the 50 to 1 µg/mL range with regression values of 0.99.

Tobramycin release was measured using the same fluorescence assay. Calibration curves were linear in the 0–25 µg/mL range with regression values of 0.99. 

Vancomycin cannot be assayed using the fluorescence assay so traditional HPLC (Waters systems, Mississauga, ON, Canada) methods were used. Vancomycin drug release was analyzed using reverse phase HPLC methods with a Waters HPLC system and a C18 Novapak column, detection at 256 nm and a mobile phase of 90% KH_2_PO_4_ buffer (10 mM) pH 3 at a flow rate of 1 mL/min. Calibration was linear in the 0–100 µg/mL range with a regression value of 0.99. 

Silver release was measured using an inductively coupled plasma (ICP) machine (Agilent, Lexington, MA., USA) under argon with a detection limit of 10 ng/mL. Calibration standards were linear in the 10 to 2000 ng/mL range. Silver standards in that range were run every 10 samples but the machine held calibration over 4 h use without the need to use multiple calibration gradients. Early time point release media were diluted 50 times to avoid silver values outside the calibration curve.

### 2.3. Bacterial Studies

#### 2.3.1. Preparation of Bacterial Cultures

*S. aureus Xen36* and *P. aeruginosa PA01* were cultured for a day in Luria Bertani (LB) broth from frozen stocks (using 20 μL bacterial sample in 20 mL (LB) broth). This was followed by a sub-culture procedure (using 200 μL in 10–20 mL LB broth) to achieve an optical density of 0.4 ed (at λ = 600 nm with an Eppendorf BioPhotometer (Eppendorf, Mississauga, ON, Canada), to give a bacterial suspension which was used as outlined below.

#### 2.3.2. Drug-Bacterial Studies

Silver nitrate or gentamicin stock solutions were initially diluted into the 0.25–256 µg/mL range and tested against *S. aureus* or *P. aeruginosa* directly. Drug studies were performed using the micro broth dilution method as recommended by the guidelines of the Clinical and Laboratory standard institute [32] with the modification of using a shorter 4 h time point. Aliquots were taken after 4 h and bacterial counts were obtained as colony forming units (CFU)/mL of serially diluted samples using visual counting of units on agar plates. Once the minimal concentrations were determined for the agents individually, 0.25 µg/mL of gentamicin and 0.1 µg/mL for silver were chosen as the fixed concentration for the respective drugs for combination studies, with ranges for the other drug being silver at 0–1 µg/mL (silver) or 0–0.1 µg/mL (gentamicin). The bacterial work was split into two sections. Initially the objective was to look for any synergistic activity of gentamicin and silver. This requires selecting a fixed % value for bacterial killing under each condition. To do this drug work with bacteria a value for bacterial killing that is easily measured was fixed and we chose a value for MIC of 95% bacterial death in four hours.

In the second section, we studied the antibacterial effect of release media. This is relevant to orthopedic scenarios where 100% bacterial death is essential. Therefore, we completed the analysis using an MIC_100_ value as this is a value orthopedic surgeons would be interested in. In all these bacterial studies triplicate incubations were set up per condition not triplicate assays from the same sample.

#### 2.3.3. Drug–Bacterial Studies Using Drug Release Media

For bacterial studies using release media from day 7, the media was stored frozen and then diluted serially to give a series of solutions with known drug concentrations and antibacterial activity against *S. aureus* and *P. aeruginosa* was assessed as above. Day 7 was chosen for media testing as the drug concentrations in the release media at that time were low but relevant. It is important to investigate whether the released drug combinations work synergistically at longer time points because that is one objective of the study, i.e., to have a more durable antibacterial response. These new studies investigate whether the drug (e.g., gentamicin or silver) has the same activity as freshly made drug because the CFU vs. drug concentration curves should approximately match. For combination studies, there is no control over the amount of each drug in release media because they release with different profiles. However, the studies do allow the ability to compare the antibacterial effect of similarly diluted samples. For example, if a 50× dilution of the silver plus gentamicin release media kills 99% of bacteria but a 6× dilution of each drug alone provides the same level of antibacterial effect then clearly the combination of drugs in bone cement offers a powerful drug recipe. Additionally, in the same example the actual concentration of silver and gentamicin is known in the ×50 dilution (as it is in the ×6 for each drug alone) so that drug concentration comparisons can also be made. These studies were initially run for gentamicin and silver and then extended to vancomycin and tobramycin. 

## 3. Results

Silver nitrate inhibited the growth of the Gram-positive *S. aureus* in a concentration dependent manner as shown in Figure 1a. At 32 µg/mL full bactericidal effects were observed. Gentamicin had a similar concentration dependent inhibitory effect as shown in Figure 1b. The Gram-negative *P. aeruginosa* was shown to be more sensitive to both drugs, with complete bactericidal activity at 2 µg/mL, as shown in Figure 1c,d, respectively. The MIC_95_ (concentration of silver nitrate or gentamicin that inhibits 95% of bacterial growth) for both agents against both bacteria are shown in Table 1.

For combination experiments with *P. aeruginosa*, the fixed concentration for gentamicin used was 0.1 µg/mL (no inhibitory effect as shown in Figure 1d) and inhibition of bacterial growth was observed at silver concentrations including and above 0.03 µg/mL (Figure 2), with a concentration killing effect up to a final concentration of 0.1 µg/mL. The MIC_95_ value (dotted red line) for this combination was 0.1 µg/mL gentamicin/0.035 µg/mL silver.

Using gentamicin at a fixed concentration of 0.25µg/mL (no inhibitory effect on *S. aureus*), little killing was observed at silver concentrations below 0.4 µg/mL, with concentration dependent inhibition of bacterial growth observed at concentrations between 0.4 µg/mL and 1 µg/mL (Figure 3). MIC_95_ (dotted red line) for this combination of silver and gentamicin was found to be at 0.25 µg/mL gentamicin/0.5 µg/mL silver.

The minimal inhibitory concentration values for gentamicin, silver and combinations of agents against *S. aureus* and *P. aeruginosa* are shown in Table 1. Much lower concentrations of either drug when used in combination was needed to reach the MIC than when using each drug individually.

When studying the release profiles of silver from bone cement, a burst phase of release over two days was observed followed by a slower release after that up to 10 days (Figure 4). The addition of dextran at 18% loading (*w/w* to cement) resulted in a much larger burst phase of release in the first hour (60%) to 5 h (73%) followed by minor levels of release after that. The release rate of silver from films co-loaded with gentamicin was lower than from films loaded with silver alone (Figure 4).

Gentamicin release profiles from bone cement was similar to that of silver, with a burst phase in the first 2 days followed by a steady release over the next 8 days (Figure 5). The release of gentamicin from dextran-loaded films was characterized by almost 100% release at 5 h with no release after that (Figure 5). In experiments not involving dextran, 60% of the silver was released over 2 days (Figure 6). The release rates were similar between films loaded with silver alone or silver plus gentamicin unlike the results observed in experiments involving dextran where the release rates were lower.

Gentamicin release from bone cement films in the absence of dextran containing either gentamicin alone or gentamicin combined with silver (Figure 7) was similar to the profiles obtained in experiments where dextran was included, whereby burst release was observed over 2 days with a low level of release after that with 100% release achieved at 10 days.

In order to assess the antibiotic potential of drug loaded bone cement over time, the release media from day 7 was tested against the Gram-positive *S. aureus* and Gram-negative *P. aeruginosa*. Because the concentration of drug in these release media (from day 4 to 7) was known, they were diluted to the same degree to allow known concentrations of released drug combinations to be assessed. The dilutions of the release media and the final concentrations of all agents is shown in Table 2. The investigation of the antibiotic effect of the release media by studying the equally diluted samples allows for the analysis of the relative antibiotic effect of drugs alone versus the combination of two. This is because as the samples get diluted, the drug concentration drops to a point at which there is little antibiotic effect.

The inhibitory effect of diluted silver or gentamicin release media (expressed as the final concentration of silver or gentamicin) is shown in Figure 8 a,b, respectively. These data closely approximate similar data for dissolved silver, as shown in Figure 1 a,b with inhibition of bacterial growth increasing dramatically at concentrations above 4 µg/mL and gentamicin showing very powerful inhibitory effects below 1 µg/mL. For dilutions of day 7 release media from the silver and gentamicin combinations, strong inhibition of bacterial growth began at combined combinations of 0.09 and 0.23 µg/mL (respectively) with almost complete inhibition at twice these concentrations. These data confirm superior antibacterial activity of combination silver/gentamicin at lower levels than those of individual drugs in Figure 2.

For *P. aeruginosa*, the inhibition of growth by media from silver or gentamicin containing bone cement films alone is shown in Figure 9 a,b, respectively. These data closely match the data using dissolved drug in Figure 1 c,d, respectively. When release media from bone cement containing both silver and gentamicin were diluted and investigated the same way, a clearly improved inhibitory effect was observed with almost full inhibition occurring at combined concentrations of 0.09 µg/mL silver and 0.23 µg/mL gentamicin as compared to full inhibition with either drug alone being achieved at much greater concentration of 1 µg/mL.

A summary of release media dilutions, silver or gentamicin concentrations, or combinations in release media along with MIC_100_ values are given in Table 3.

The data clearly show a potentiation or synergistic antibacterial effect far in excess of a simple additive effect.

Whilst gentamicin is a broad-spectrum antibiotic, vancomycin is really only effective against Gram-positive bacteria and although tobramycin has some effect against Gram-positive, the drug has greater activity than gentamicin against Gram-negative bacteria. These drugs were included in bone cement samples in order to investigate release profiles and possible improved combination antibiotic effects with silver against *S. aureus* or *P. aeruginosa.* Both drugs released from bone cement films with profiles similar to gentamicin with 2 day burst release effects and low levels of release after that (Figure 10 and Figure 11). These profiles are similar to those observed for gentamicin (Figure 7) except the final amount of drug released by day 10 was approximately 100% for gentamicin but only 70% for vancomycin or 30% for tobramycin.

The release media for day 7 (i.e., released between day 4 and day 7) from these studies was then serially diluted and the antibacterial effect tested against both bacterial species. The seven-day release media from films containing silver alone demonstrated powerful antibacterial effects against *P. aeruginosa* up to approximately 25× dilution levels (Figure 12a). On the other hand, the release media from tobramycin only proved inhibitory when used in the undiluted state (Figure 12c). However, when tobramycin and silver were used in combination at much lower concentrations than each drug alone the undiluted media killed all bacteria (Figure 12b), demonstrating an improved antibacterial effect over the use of each agent alone.

The 7-day release media from bone cement loaded with vancomycin alone had no effect on the growth of *P. aeruginosa* (Figure 13c). Inhibitory effects of release media from silver alone for these experiments shown in Figure 13a is the same as the data in Figure 12a. However, 7-day release media from bone cement samples containing both silver and vancomycin demonstrated full bactericidal activity using 1× or 3× undiluted samples (Figure 12b). Silver had a moderate antibiotic effect against *S. aureus* with 1×, 3× or 6× dilutions inhibiting bacterial growth (Figure 14a). The release media from bone cement containing tobramycin alone had no inhibitory effect (Figure 14c) and the release media containing both agents (Figure 14b) had similar activity profiles to silver alone. Similar results to tobramycin were obtained for silver alone (Figure 15a) and vancomycin alone (Figure 15c) with *S. aureus* except that the undiluted media from vancomycin showed some inhibitory effect. Otherwise there was a mild improved combination antibacterial effect observed for the 12× diluted release media (Figure 15b) but otherwise no additive effect of having both agents in bone cement.

## 4. Discussion

It has now been over 30 years since the first use of gentamicin in bone cement and the successful treatment of bacterial infections using a local controlled release formulation. Bone cement (as poly methyl methacrylate) was selected as the polymer for this purpose solely because the material was known to be biocompatible with orthopedic sites, not because it was suited to effectively deliver drugs over extended periods [6]. In fact, the polymer excludes water and effectively prevents deeper deposits of drug from moving to and releasing from the surface [20]. These poor release profiles and the subsequent development of drug resistant bacteria to gentamicin [2,5,7] and other relevant antibiotics have seriously impacted the effectiveness of this type of treatment. Furthermore, while much focus is on the development of new antibiotics or other novel antibiotic strategies to overcome mainly resistance issues, logistical and ethical dilemmas exist around testing new drugs in seriously infected patients including those suffering from serious orthopedic infections [33]. As a result, this study investigated a strategy of using combinations of antibiotic agents in bone cement in an attempt to improve the antibiotic action via synergistic effects that might allow the killing of bacteria at much lower concentrations of drugs and minimize the development of resistance than when single drugs are used alone. The MIC values for gentamicin ranged between 0.25 to 0.5 µg/mL for Gram-negative bacteria and 0.5–4 µg/mL range for Gram-positive bacteria are close to those reported in other studies [34,35]. The MIC values for silver nitrate in the 0.25 to 4 µg/mL range for Gram-negative and -positive bacteria (respectively) are lower than those reported by others (5 to 13 µg/mL) [26,36]. However, when gentamicin was used at concentrations that had no effect on the growth of either *S. aureus* or *P. aeruginosa* the addition of silver nitrate almost fully inhibited bacterial growth at concentrations of just 0.4 µg/mL or 0.03 µg/mL, respectively (Figure 2 and Figure 3). Clearly these combined effective concentrations point to a powerful synergistic activity, a phenomenon that has previously been reported for gentamicin and other antibiotics [34,37,38,39,40,41,42]. Interestingly, the data might suggest an alternative selection of silver over gentamicin as a single agent for inclusion in bone cement. However despite the low toxicity of silver in the body [43] it is unlikely that orthopedic surgeons would be prepared to drop the use of well-established antibiotics like gentamicin, tobramycin and vancomycin. It might be more acceptable to promote the synergistic aspect of using two drugs as this allows for lower drug loadings and extended antibacterial actions.

We have previously shown that the addition of sodium chloride or dextran up to 18% in bone cement has no significant effect on the mechanical strength but greatly enhanced antibiotic release [20]. In the present study we chose to use only dextran as an additive, as the presence of sodium ions with the use of NaCl would have compromised silver detection by ICP methods [44]. Furthermore, while both beads and films were used in a previous study [20] only films were used in the present studies as the geometry allows for a greater surface area to volume ratio, increasing drug release and the ability to better investigate the antibiotic effects of released drug at later time points. The addition of dextran to bone cement caused a large increase in the burst phase of drug release for both silver and gentamicin, as shown in Figure 4 and Figure 5, in agreement with previous work involving vancomycin, fusidic acid and linezolid. However, because the inclusion of dextran resulted in very low amounts of silver or gentamicin release at later time points (Figure 4 and Figure 5) further studies did not include dextran. Generally, both gentamicin and silver release profiles were similar to those reported previously by our group [20] and others whereby most of the drug was released in the first few days with only small amounts released after that (Figure 4, Figure 5, Figure 6 and Figure 7) [8,13,14,15,16]. Whilst the high levels of drug release in the first two days would be bactericidal it is likely that the small amounts of drug released after that time might not be effective and very low concentrations could induce drug resistance [17] warranting future studies to investigate modifications that result in later-stage release profiles at bactericidal concentrations.

The important and central aspect to this work was to investigate whether the drug release media from later time points (day 7) had antibiotic effects against *S. aureus* and *P. aeruginosa*. The dilutions were converted to concentrations of either silver or gentamicin and the antibacterial effects (Figure 8 a,b for *S Aureus*) (Figure 9 a,b for *P aeruginosa*) were similar to those shown in Figure 1a–d (using dissolved drug) confirming the quantitative nature of the gentamicin and silver assays and demonstrating little drug degradation over the 7-day time frame which is important for in vivo applications. However, the release media from bone cement containing both agents had a much more powerful antibiotic effect than release media from bone cement with single agents (Figure 8c and Figure 9c) where silver and gentamicin concentrations of 0.36 + 0.92 µg/mL (*S. aureus*) or 0.18 + 0.46 µg/mL (*P. aeruginosa*) were clearly synergistic in effect. The lower values of both drugs for *P. aeruginosa* reflect the lower MICs for both drugs individually for these bacteria as compared to *S. aureus* (Table 1).

This work suggests two options for improved antibiotic treatments using bone cement. One is to use less antibiotics in the bone cement by leveraging the synergistic action of two agents and the other is to extend the duration of efficacy by allowing existing slow drug release profiles to give combined synergistic concentrations of drugs whereas single drugs might be ineffective.

Other groups have loaded silver nanoparticles and gentamicin in bone cement or polycaprolactone implants and found no or only mild additive antibacterial effects [24,45,46]. However, Zhou et al. [38] coated orthopedic implants with silver nanoparticles and gentamicin coatings and demonstrated good antibiotic synergy.

In the field of wound dressings, the use of silver nanoparticles dominates over the use of silver salts due to silver ion staining of tissues and a proposed increased activity of nanoparticles. However wound dressings are generally permeable to water so that silver nanoparticles can effectively release to the active tissue area. Of course, in orthopedic settings silver staining is not as critical an issue as in dermal applications. In bone cement it is likely that the release of insoluble silver nanoparticles is even more restricted than that of freely soluble silver salts potentially explaining the non-compelling nature of studies so far using gentamicin and silver nanoparticles in bone cement. There is also a perceived but unproven toxicity risk of using silver salts in bone cement based on early studies where some patients with longer residing bone cement had poor outcomes with cement loaded with high amounts of silver [21,22,26]. Clearly these issues direct caution in the use of silver salt loaded bone cement. However, the use of low concentrations of silver salts not generally associated with negative effects in combination with gentamicin may warrant the use especially in areas where a removable spacer or anti-infective beads are used for a short time. In such cases, a choice between the elimination of potentially life-threatening bacteria against the risk of potential side effects from silver toxicity might be justified.

In light of the much-improved antibiotic performance arising from combining silver nitrate and gentamicin in bone cement, these studies were extended to similar studies using tobramycin and vancomycin. Bone cement was loaded with these drugs alone (0.8% *w/w*) or in combination with silver nitrate (0.8% *w/w*) and drug release studies were run. Release study profiles were similar to gentamicin. The release media from the 7-day sample was then serial diluted (like gentamicin) and antibacterial effects measured. As expected the release media for silver alone was effective against both bacteria (similar to gentamicin studies) but surprisingly the release media from tobramycin had no antibiotic effect. This may have resulted from the very low levels of drug release shown for Tobramycin (Figure 11). The lack of an enhanced effect from combining with silver also probably arises from the low levels of tobramycin released between day 4 and day 7. Similar results were found for vancomycin with *S. aureus* which may also have arisen from the low levels of drug released from cement in the 4–7-day time period. There were greatly enhanced combination effects for both vancomycin and tobramycin against Gram-negative *P. aeruginosa* with 100% bactericidal activity observed using the undiluted release media despite the low levels of drug released.

Tobramycin has good activity against Gram-negative bacteria and it has been previously reported that silver nanoparticles enhance the antibacterial effect of this drug against *P. aeruginosa* [47]. Similarly, silver has been reported to sensitize Gram-negative bacteria to vancomycin [26].

Overall these data suggest that for gentamicin there is a clear advantage of including low levels of silver salts in bone cement to extend or strengthen the antibiotic activity of the cement in the presence of both Gram-positive and Gram-negative bacteria. This advantage arising from the clear synergistic action of the combination against both types of bacteria. There is also evidence that there is a similar advantage for both tobramycin and vancomycin in the presence of Gram-negative bacteria but poor longer-term drug release profiles may argue against combined use against Gram-positive bacteria.

## Figures and Tables

**Figure 1 materials-14-03413-f001:**
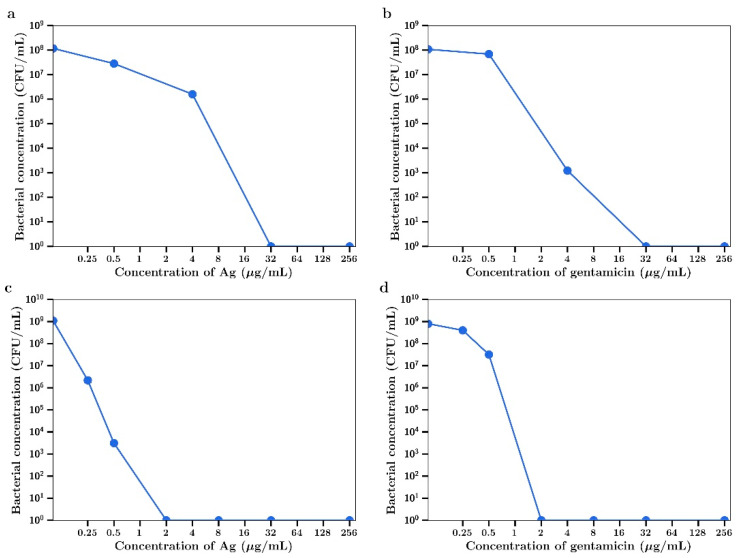
Bacterial concentration at t = 4 h for (**a**) AgNO_3_ against *S. aureus*, (**b**) gentamicin against *S. aureus,* (**c**) AgNO_3_ against *P. aeruginosa*, (**d**) gentamicin against *P. aeruginosa*.

**Figure 2 materials-14-03413-f002:**
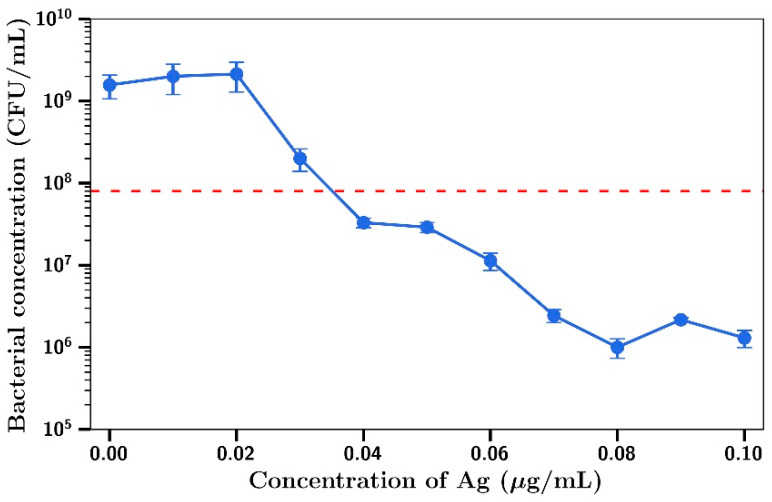
*P. aeruginosa* concentration at t = 4 h with 0.1 µg/mL gentamicin and varying Ag concentrations (*n* = 3).

**Figure 3 materials-14-03413-f003:**
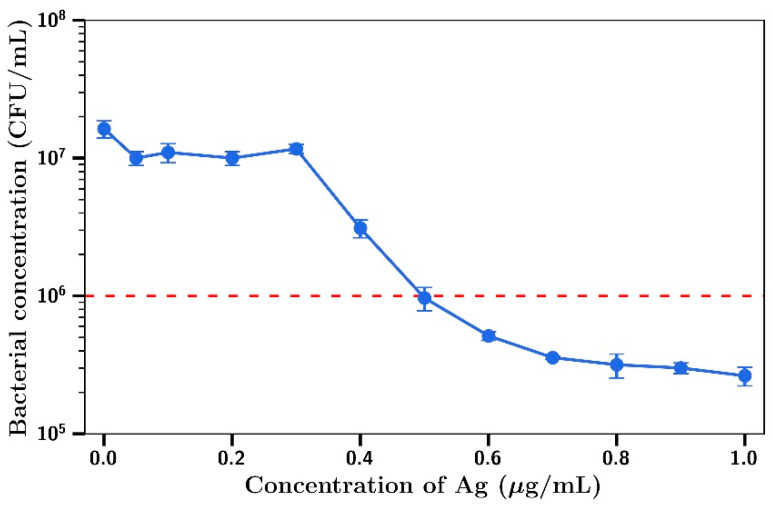
*S. aureus* concentration at t = 4 h with 0.25 µg/mL gentamicin and varying Ag concentrations (*n* = 3).

**Figure 4 materials-14-03413-f004:**
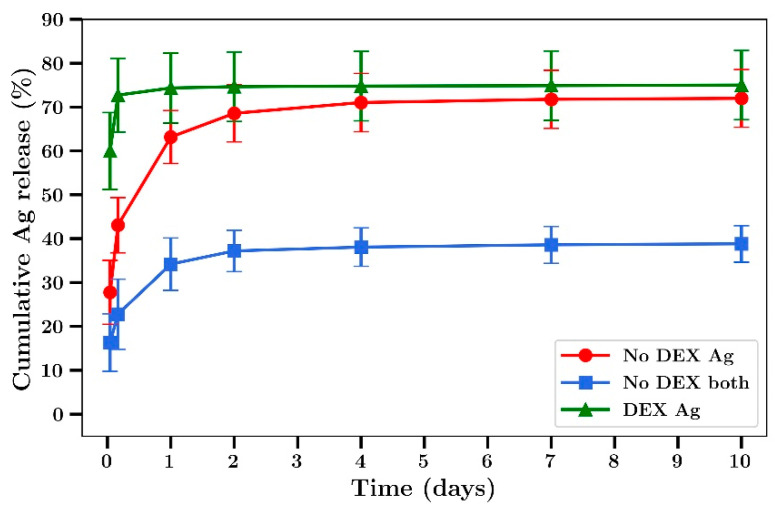
Time course of Ag release from Palacos bone cement films. Cement loaded with or without dextran T70 (18% *w/w*) and silver nitrate. (Silver loaded at 0.8% and gentamicin at 0.8% *w/w*). No DEX both refers to cement with gentamicin and Ag and no dextran (*n* = 4).

**Figure 5 materials-14-03413-f005:**
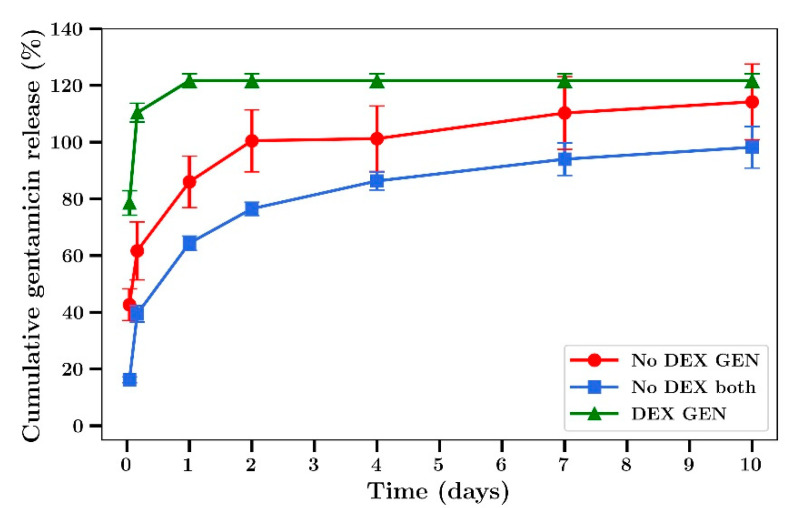
Time course of gentamicin release from Palacos bone cement films. Cement loaded with or without dextran T70 18% *w/w*) and gentamicin. (Silver loaded at 0.8% and gentamicin at 0.8% *w/w*) No DEX both refers to cement with gentamicin and Ag and no dextran (*n* = 4).

**Figure 6 materials-14-03413-f006:**
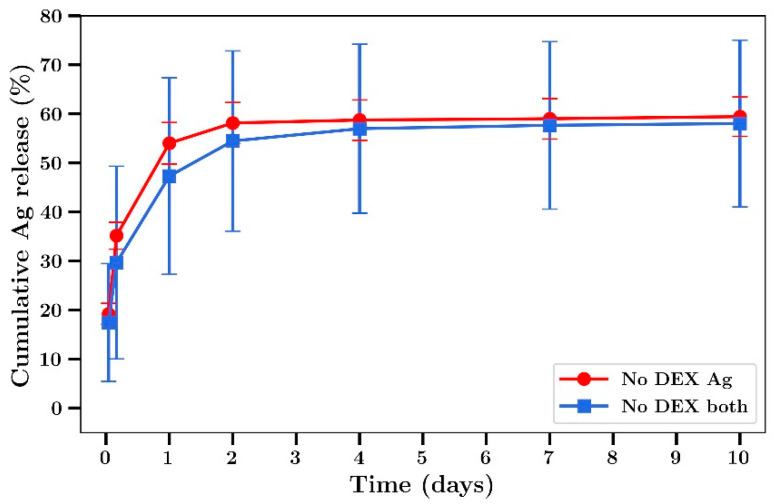
Time course of Ag release from Palacos bone cement films (no dextran) loaded with Ag alone or Ag plus gentamicin. (Silver loaded at 0.8% and gentamicin at 0.8% *w/w*) (*n* = 4).

**Figure 7 materials-14-03413-f007:**
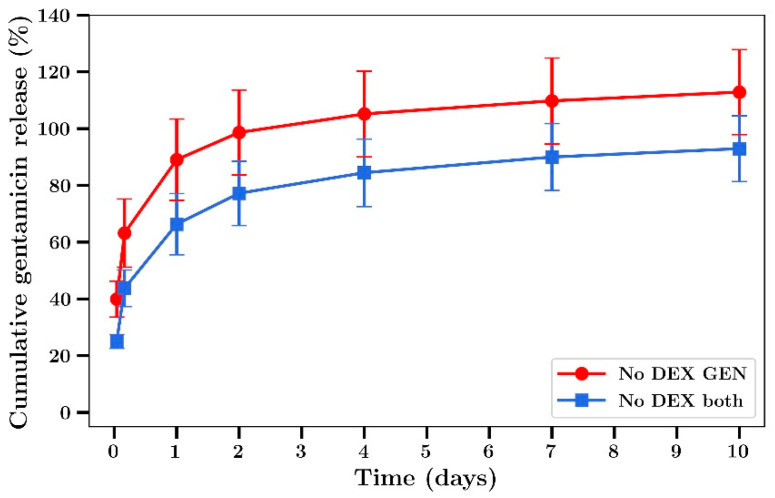
Time course of gentamicin release from Palacos bone cement films (no dextran) loaded with gentamicin alone or gentamicin plus Ag. (Silver loaded at 0.8% and gentamicin at 0.8% *w/w*) (*n* = 4).

**Figure 8 materials-14-03413-f008:**
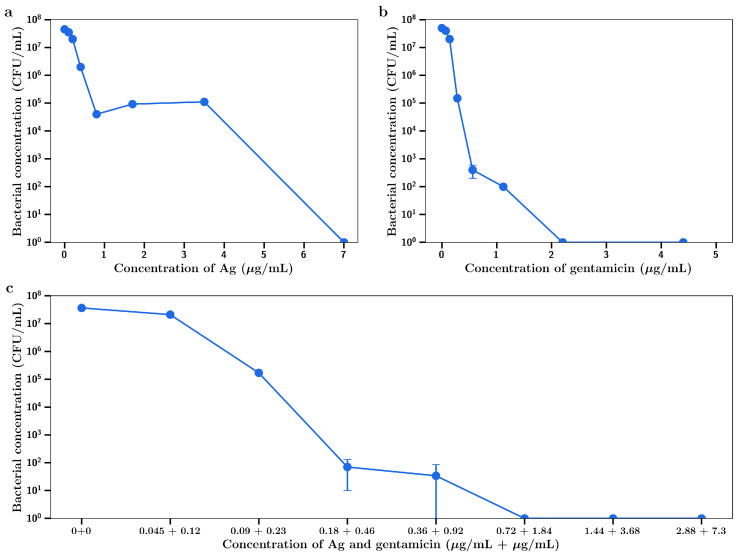
Bacterial concentrations of *S. aureus* incubated in day 7 release media from (**a**) Ag-releasing bone cement, (**b**) gentamicin-releasing bone cement, (**c**) Ag + gentamicin-releasing bone cement (*n* = 3).

**Figure 9 materials-14-03413-f009:**
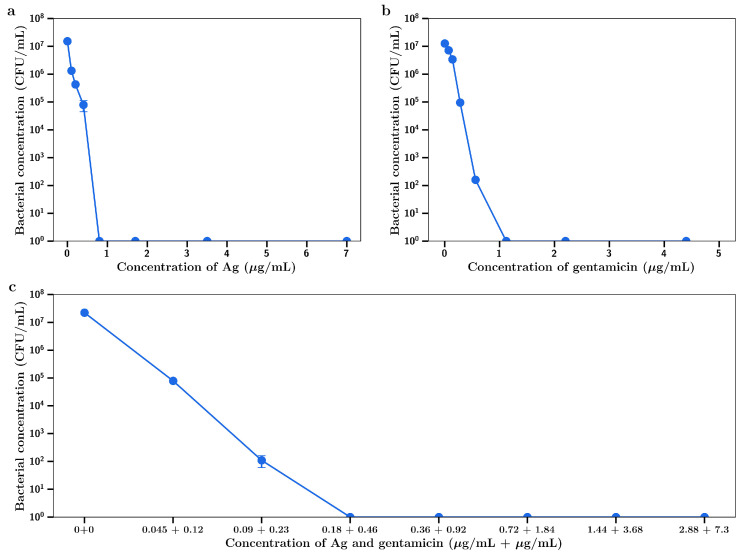
Bacterial concentrations of *P. aeruginosa* incubated with day 7 release media from (**a**) Ag-releasing bone cement, (**b**) gentamicin-releasing bone cement, (**c**) Ag + gentamicin-releasing bone cement (*n* = 3).

**Figure 10 materials-14-03413-f010:**
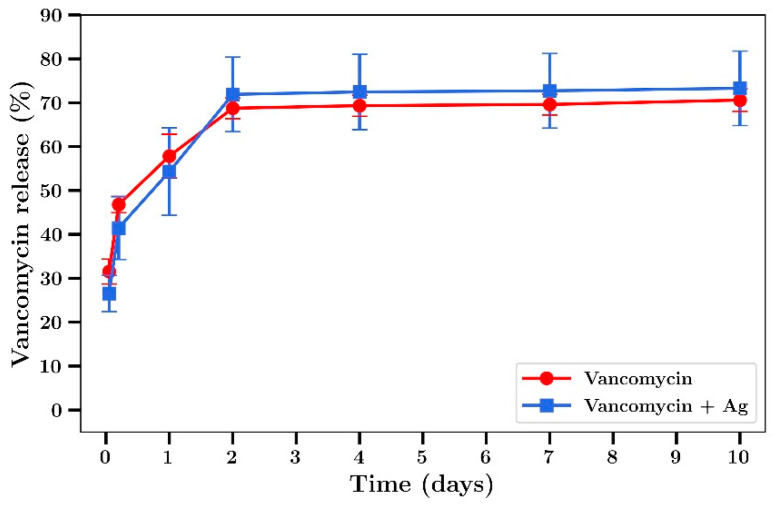
Time course of release of vancomycin from bone cement films containing vancomycin alone or vancomycin with Ag. (Silver loaded at 0.8% and vancomycin at 0.8% *w/w*) (*n* = 4).

**Figure 11 materials-14-03413-f011:**
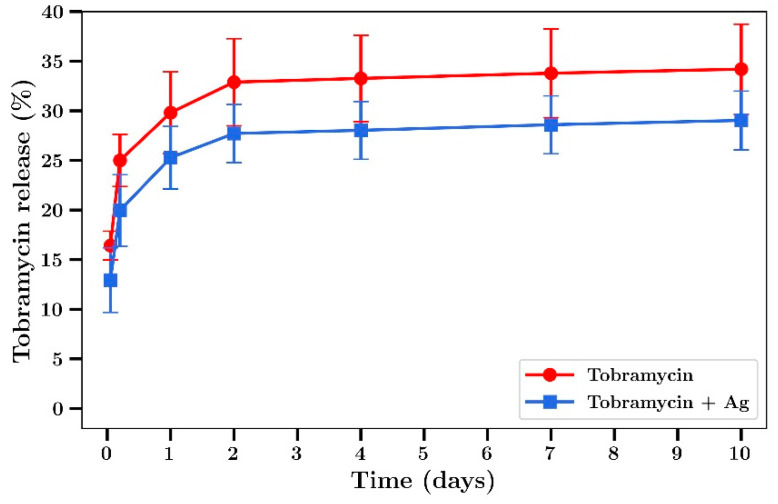
Time course of release of tobramycin from bone cement films containing tobramycin alone or tobramycin plus Ag. (Silver loaded at 0.8% and tobramycin at 0.8% *w/w*) (*n* = 4).

**Figure 12 materials-14-03413-f012:**
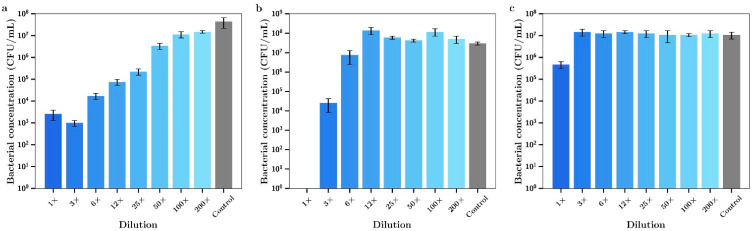
Bacterial concentrations of *P. aeruginosa* incubated in various dilutions of the 7-day incubation release media of bone cement containing (**a**) Ag, (**b**) Ag + tobramycin, (**c**) tobramycin. (Silver loaded at 0.8% and tobramycin at 0.8% *w/w*) (*n* = 3).

**Figure 13 materials-14-03413-f013:**
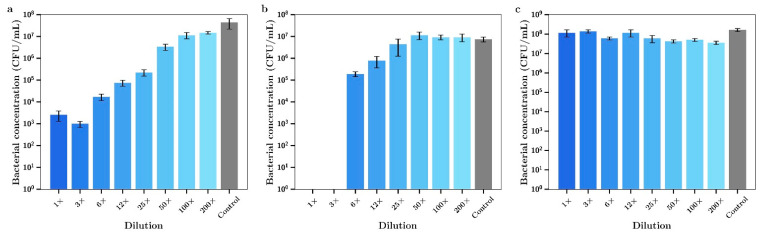
Bacterial concentrations of *P. aeruginosa* incubated in various dilutions of the 7-day incubation release media of bone cement containing (**a**) Ag, (**b**) Ag + vancomycin, (**c**) vancomycin. (Silver loaded at 0.8% and vancomycin at 0.8% *w/w*) (*n* = 3).

**Figure 14 materials-14-03413-f014:**
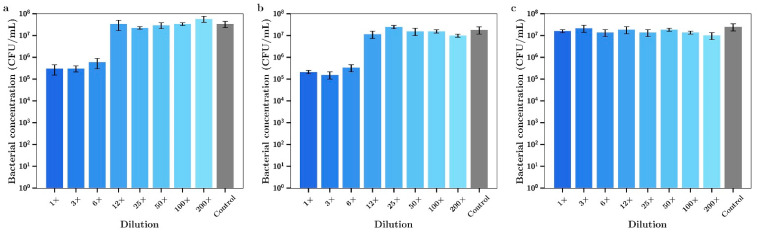
Bacterial concentrations of *S. aureus* incubated in various dilutions of the 7-day incubation release media of bone cement containing (**a**) Ag, (**b**) Ag + tobramycin, (**c**) tobramycin. (Silver loaded at 0.8% and tobramycin at 0.8% *w/w*) (*n* = 3).

**Figure 15 materials-14-03413-f015:**
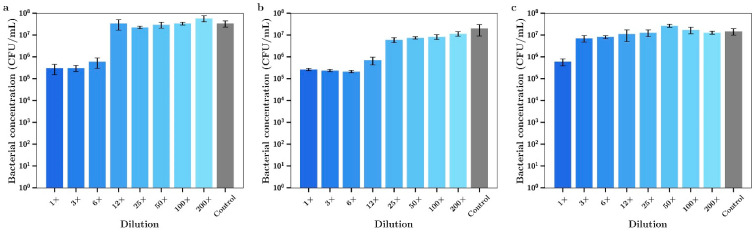
Bacterial concentrations of *S. aureus* incubated in various dilutions of 7-day incubation release media from bone cement loaded with (**a**) Ag, (**b**) Ag + vancomycin, (**c**) vancomycin. (Silver loaded at 0.8% and vancomycin at 0.8% *w/w*) (*n* = 3).

**Table 1 materials-14-03413-t001:** MIC_95_ values for silver nitrate, gentamicin or combinations against *P. aeruginosa* and *S. aureus.*

Bacteria	MIC_95_		Combined MIC_95_	
	Silver Nitrate (µg/mL)	Gentamicin (µg/mL)	Silver Nitrate (µg/mL)	Gentamicin (µg/mL)
*P. aeruginosa*	0–0.25	0.25–0.5	0.03–0.04	0.1
*S. aureus*	0.5–4	0.5–4	0.4–0.5	0.25

**Table 2 materials-14-03413-t002:** Concentration of drugs in day 7 diluted release media from bone cement loaded with silver nitrate, gentamicin or both.

Dilution	Ag	Gentamicin	Ag + Gentamicin	
	Concentration of Ag (µg/mL)	Concentration of Gentamicin (µg/mL)	Concentration of Ag (µg/mL)	Concentration of Gentamicin (µg/mL)
3×	7	4.4	2.88	7.3
6×	3.5	2.2	1.44	3.68
12.5×	1.7	1.12	0.72	1.84
25×	0.8	0.56	0.36	0.92
50×	0.4	0.28	0.18	0.46
100×	0.2	0.14	0.09	0.23
200×	0.1	0.07	0.045	0.12

**Table 3 materials-14-03413-t003:** MIC_100_ values obtained from day 7 drug release media for bone cement loaded with silver nitrate, gentamicin or both. Table shows the degree of dilution necessary for the release media to kill the bacteria as well as the concentration of drug(s) in that release media.

Drug Dilution	*S. aureus*			*P. aeruginosa*		
	Ag	Gentamicin	Both	Ag	Gentamicin	Both
Dilution	6–3×	12–6×	25–12×	50–25×	25–12×	100–50×
Concentration (µg/mL)	3.5–7	1.12–2.2	Ag: 0.36–0.72 GEN: 0.92–1.84	0.4–0.8	0.56–1.12	Ag: 0.09–0.18 GEN: 0.23–0.46

## Data Availability

Not applicable.

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
