# Peer review of "The Combined Use of Gentamicin and Silver Nitrate in Bone Cement for a Synergistic and Extended Antibiotic Action against Gram-Positive and Gram-Negative Bacteria"

_materials, 2021, doi:10.3390/ma14123413_

Round 1

Reviewer 1 Report

This is an interesting work with a potential clinical application

Most of the comments that I presented in the first revision process were addressed by the authors. However, I still have some questions.

Specific comments:

  1. Materials and methods section:
    1. In the bacterial studies the authors referred the application of the Clinical and Laboratory standard institute (CLSI) guidelines. However, CLSI do not refer MIC determinations after 4 hours of incubation. This point must be clarified. You can say in the manuscript that a modified version of the CLSI guidelines was used for MIC determination.
    2. The antimicrobial concentration values used for MIC determination described by the authors were 1 to 100 µg/µL. Considering that, how the authors refer in the drug-bacterial studies and results section MIC values of 0.25 or 0.5 µg/mL?;
    3. Lines 33 to 43 must be clarified. The justification presented its messy and you should include some reference in it.
    4. Finally, it is important to perform this kind of experiments at least 3 times in independent assays, to allow results reproducibility. I notice that now, some figures, refer the number of assays. However, the information regarding the number of experiments performed should be included in the methods section.
  2. Results section:
    1. In the table 1 the results should be presented as mean ± standard deviation. The range may be an option if you have different strains in each row. In this case only one aureus and one P. aeruginosa were evaluated.
    2. All figures have an antimicrobial range of 0 to 256µg/mL, when it is described in the methods that a range of 1 to 100 µg/mL was used. Please clarify this.
    3. In lines 20-21 you mentioned the release of gentamicin from dextran-loaded in Figure 4. This must be corrected because the Figure that refer the Gentamicin release it is not figure 4 and it is in the manuscript without any legend.
    4. Figure 5 have an error in the Y axis. It should be cumulative gentamicin release and not “Cumulative Ag release (%)”.

Reviewer 2 Report

The topic is interesting even though the presentation of the results is not accurate. Authors need to check and unify the data that is in the figures and unify it with what they write in the text. In addition, there are many typos and errors in the text. The article gives the impression that the authors did not pay enough attention to it. In addition, there is no conclusion, no summary of the results. Questions and comments on the article:

  1. How was MIC95 calculated?
  2. Please control the results in paragraph page 6 lines 227-236and Figs. 2, 3.
  3. Authors should review the results described in the text with the results in the graphs. Sometimes they differ considerably.
  4. Figure 3. P.  aeruginosa concentration at t=4h with 0.25 μg/mL gentamicin and varying Ag concentrations. (n=3). But in the text, the concentration is 0,1.
  5. Why the curves of „No DEX both“ in Figures 4 and 5 are not the same?
  1. Page 15 lines 449 452 - How do the authors explain these differences in findings?
  2. There is a lot of formal mistakes in the whole text, for instance:
    1. Page 3 lines 116 and 123 – 1.5 g and 19mg – sometimes the authors put a space after the number, other times they don't. It happens through the text. Please correct
    2. Many unnecessary spaces between words - Page 3 line 132; Page 4 lines 148, 156, 171, 173, 181, etc. Please correct
    3. Page 8 line 226; Parenthesis is missing

Author Response

see attachment

This manuscript is a resubmission of an earlier submission. The following is a list of the peer review reports and author responses from that submission.

Round 1

Reviewer 1 Report

Line numbering and section numbering: Needed for tracking of corrections, please do the needful.

Materials: Please include every single material that was used in the experimentation.

Bone Cement Film Manufacture: Please rewrite the section. It is confusing the way it is presented. May you want to add a schematic for the process.

Percent drug loading: Please include results of drug loading. Without that the drug release study has no value.

Drug Release Study: Please rewrite the section. It is very distorted in the way it presented. You may want to add another section for the method of analysis.

Bacterial Studies: The same comment applies here as a ‘drug release study.' Please do the needful.

Figure 1: Caption Missing.

Figure 1: Check the axis labels.

There are no captions for any of the figures. How to read understand the figure?? Please do the needful.

Reviewer 2 Report

Comments to the authors:

The main goal of this work was to investigate the antimicrobial activity of several compounds (silver nitrate, gentamicin, vancomycin and tobramycin, as well as the interaction between silver nitrate and the antibiotics) incorporated in bone cement or bone cement with dextran, on S. aureus and P. aeruginosa. Also, the release capacity of bone cement with dextran was also evaluated.

This is a relevant study for clinical practice that provide some information to the current knowledge.

Specific comments:

  1. Introduction section: in the first paragraph it is necessary to include the references that supports all sentences. Only one reference in one sentence is presented. In the 5th paragraph the same occurs in the first two sentences.
    1. In the last paragraph we can observe the goals of the study. However, no description about the use of vancomycin and tobramycin is mentioned, please clarify that.

  1. Materials and methods section: In this section major revisions should be performed.
    1. In bone cement film manufacture: the paragraph should not be initiated by a number. The paragraph should include references. The concentration of the silver nitrate and antibiotics incorporated in the bone cement must be included, this is a critical point. In the 3rt line of the paragraph “…(25mm x75mm x 1mm) and a second microscope was applied” correct to “…(25mm x75mm x 1mm) and a second slide was applied…”.
    2. In drug release studies: Timepoints used for the sample collection should be included and not “At appropriate timepoints…”. Why was day 7 chosen for all the evaluation? References are missing in all topic. The concentration units are misspelling, please correct over the manuscript “ug/ml” to µg/mL. Why vancomycin release was measure with a different methodology, when compared with the other antimicrobials?
    3. In the bacterial studies: it must be clarified and presented the methodology used to determine the minimum inhibitory concentrations (MICs). It is not clear if the method was broth microdilution technique, and if so, why samples were collected after 4h of exposure? What is the reference of this MIC determination? I advise the authors to see the guidelines of the Clinical and Laboratory standard institute (CLSI) where its is described MIC methodologies.
    4. It is missing a topic about the statistical analysis performed. In the results section it is mentioned some statistical analyses that allow the choose of the bone cement without dextran for example (mentioned after figure 4).
    5. Finally, it is critical for the manuscript to perform this kind of experiments at least 3 times in independent assays, to allow results reproducibility. Please present the results of 3 independent assays with means ± standard deviation.

  1. Results section: All Figures must include a title and a legend. Table 1 must include a legend with the abbreviation MIC explained.
    1. In the first paragraph you said “The gram-negative aeruginosa was shown to tolerate higher concentrations…” but your results said that MIC values of both antibiotics are lower for P. aeruginosa then S. aureus. Can you clarify that?
    2. Why have been used values of MIC95 and MIC100? This is a mistake? Standardization of the methodology must be performed as well as results.
    3. I think you should explain better the use of dilutions to evaluate the antimicrobial activity. I understand that the synergism effect was evaluated this way, however you should include some information about that. Do you expect to reduce the antimicrobial or silver nitrate concentration in the bone cement after this experiment? Or you only perform that to understand the interaction between compounds? If so, no difference will be observer in practice potential use of these products.
    4. In Figure 6 inside the graphic the indication “No DEX GEN” is wrong. It should be No DEX Ag.
    5. In figure 4, 5 and 11 the variable No DEX both reveals that the presence of both compounds promotes a delay in their release. Only with vancomycin the presence of Ag seemed to promote a high release. Do you know why is that happening?

  1. In the Discussion section:
    1. Include references in the first 3 sentences.
    2. Page 12, lines 17 to 19 please change “The MIC values for gentamicin in the 0.25 to 0.5 ug/ml range for gram-negative bacteria…” to “The MIC values for gentamicin ranged between 25 to 0.5 ug/ml for gram-negative bacteria…”
    3. Page 13 line 7 include reference and line 8 include the mining of ICP and a reference. Line 14 include reference.
    4. Page 14 line 12 include de figure number
    5. If the silver nitrate showed good antimicrobial activity and considering the need to reduce antimicrobial use, why do you not consider the use of bone cement only with silver nitrate? What is the cytotoxicity of this compound? Despite the reduced MIC values presented, the initial load with antimicrobials in bone cementum should be changed in a in vivo application? These points should be included and discussed in this section.